# A Multifunctional Biosurfactant Extract Obtained from Corn Steep Water as Bactericide for Agrifood Industry

**DOI:** 10.3390/foods8090410

**Published:** 2019-09-12

**Authors:** Alejandro López-Prieto, Xanel Vecino, Lorena Rodríguez-López, Ana Belén Moldes, José Manuel Cruz

**Affiliations:** 1Chemical Engineering Department, School of Industrial Engineering—Centro de Investigación Tecnológico Industrial (MTI), University of Vigo, Campus as Lagoas-Marcosende, 36310 Vigo, Spain; alexlopez@uvigo.es (A.L.-P.); lorena@uvigo.es (L.R.-L.); jmcruz@uvigo.es (J.M.C.); 2Chemical Engineering Department, Universitat Politècnica de Catalunya (UPC)—Barcelona TECH, Campus Diagonal, Besòs, 08930 Barcelona, Spain; 3Barcelona Research Center for Multiscale Science and Engineering, Universitat Politècnica de Catalunya (UPC)—Barcelona TECH; Campus Diagonal, Besòs, 08930 Barcelona, Spain

**Keywords:** corn, water stream, biosurfactant, bactericide, *Pseudomonas aeruginosa*, *Escherichia coli*

## Abstract

The increase of crop production along with stricter requirements on food security have augmented the demand of new and eco-friendly bactericides. Most of the bactericides used at the moment consist of persistent organic substances, representing a risk for environmental and human health. For instance, agriculture bactericides used for crop protection includes copper-based, dithiocarbamate and amide bactericides, which are not biodegradable, resulting in the necessity of further research about the production of new active principles that attack microorganisms without producing any harmful effect on human health or environment. The biosurfactant extract evaluated in this work as a bactericide, is obtained from corn steep water, a residual stream of corn wet milling industry, which is fermented spontaneously by probiotic lactic acid bacteria that possess the capacity to produce biosurfactants. In previous works, it has been demonstrated that this biosurfactant extract is able to promote the growth of *Lactobacillus casei* in drinkable yogurts, though its antimicrobial activity against pathogenic strains has not been evaluated at the moment. The results obtained in this work have proved that this biosurfactant extract is effective as bactericide against *Pseudomonas aeruginosa* and *Escherichia coli,* at concentrations of 1 mg/mL, opening the door to its use in agrifood formulations for reducing the use of chemical pesticides and preservatives.

## 1. Introduction

Biosurfactants are surface-active compounds of microbial origin. They are able to reduce the surface tension of water helping in the stabilization of emulsions and in the solubilization of hydrophobic substances in formulations, with further and potential applications in a wide range of areas in the industry, such as the cosmetic, pharmaceutical or food industry [1,2,3]. An increasing interest has risen in recent years where several studies have focused on the production, extraction and application of biosurfactants, which are known to be less toxic, more efficient, more biodegradable and more biocompatible than surfactants of chemical origin, due to their composition constituted of lipids, proteins and/or sugars [4,5]. Moreover, some biosurfactants possess other interesting properties, in addition to its surfactant activity. Therefore, López-Prieto et al. [3] have demonstrated that a biosurfactant extract obtained from a residual stream, of corn wet milling industry, named corn steep water (CSW) and fermented spontaneously by lactic acid bacteria, is able to promote the growth of *Lactobacillus casei* (*L. casei*) in drinkable yogurts.

Agroindustrial residues are an important source for the production and extraction of biosurfactants as it has been shown in previous works. For instance, vineyard pruning waste and other lignocellulosic residues can be used as carbon source by *Lactobacillus paracasei* (*L. paracasei*) or *Lactobacillus pentosus* (*L. pentosus*) to produce biosurfactants [6,7], most of them fermented by probiotic lactic acid bacteria. The use of CSW as a source of biosurfactant extracts with multifunctional properties has been proven in various works [5,8]. This stream is produced during the treatment of corn with SO_2_ in order to soft and swell it, in the presence of lactic acid bacteria, which are known by their probiotic activity. Therefore, Vecino et al. [9] established a liquid–liquid (L–L) extraction protocol with organic solvents to obtain a biosurfactant extract from CSW. Both chloroform and ethyl acetate can be used as organic solvents. In the case of food applications, the European Union (EU) regulation allows the use of ethyl acetate as organic solvent for L–L extractions [10]. López-Prieto et al. [3] showed that the biosurfactant extract obtained by L–L extraction with ethyl acetate, as organic solvent, could be incorporated on a drinkable yogurt helping in the development and growth of *L. casei* presented in it. Therefore, it could favour the growth of *L. casei*, a probiotic bacterium, in the intestine and its inclusion as a prebiotic component in food products.

Biosurfactants extracted from CSW have been able to reduce the surface tension of water up to 30 mN/m [9] and have been characterized as lipopeptide biosurfactants composed by C16 and C18 fatty acids. In the extract, there are also phenolic compounds that have showed antioxidant properties [2].

The production of biosurfactants by microorganisms could be related with extreme growth conditions or with the presence of pathogenic microorganisms. Hence, previous studies have highlighted the anti-adhesive properties of biosurfactants against pathogens, reducing the adhesion of bacterial pathogens to silicone rubber or voice prostheses [11] and in the removal of pathogenic biofilms [12]. Moreover, microbial surfactants produced by *Lactobacillus* spp. have been reported to show antibacterial activity against food-borne pathogens, such as *Pseudomonas aeruginosa* (*P. aeruginosa*), *Escherichia coli* (*E. coli*) and *Staphylococcus aureus* (*S. aureus*) [13,14,15,16].

Microbial food spoilage is one of the main problems that the food industry has to overcome, resulting in important economic and product losses. It is estimated that 33% of the food supply is lost due to food spoilage of microbial origin [17]. Some of the microbial contaminations found on food products are caused by pathogenic strains. Among them, *P. aeruginosa* and *E. coli* are two of the most important ones. *P. aeruginosa* spp. are usually adaptable to different conditions and can be found in a wide range of products. Their presence in processes that imply heating, like the production of ultra-high temperature (UHT) milks and dairy products, represents the main problem of contamination by *P. aeruginosa* spp. in the food industry, due to their heat-resistant characteristic that makes them difficult to remove [18]. In humans, *P. aeruginosa* is able to provoke infections and diseases in several tissues and sites [19] and it is well known for causing severe infections in immunocompromised patients [20].

*E. coli* is a Gram-negative bacterium known in the food industry for its capacity to colonize and form biofilms in surfaces [21] and for its fast antimicrobial resistance [20]. It is able to infect the gastrointestinal tract in humans, provoking several intestinal diseases as well as affecting to the pulmonary and nervous system [22].

Some of these contaminations are related with the production of vegetables, where pesticides and preservatives play an essential role in the inhibition of the growth of pathogens. The expansion of arable areas, along with the increase of crop production and new restriction for the formulations of pesticides demand the need of research on new biodegradable and eco-friendly pesticides that are harmless for humans. The European Commission issued a legislation of some persistent organic components that need to be ruled out of pesticides formulations like copper-based and amide bactericides [23], which are not biodegradable and are harmful with the environment. Therefore, biosurfactants appear as alternatives to these compounds to be introduced in agrifood formulations to reduce the use of chemical pesticides and preservatives.

The aim of this work was to evaluate the bactericide activity of a multifunctional biosurfactant extract obtained from CSW, a corn wet milling industry residue fermented spontaneously by probiotic lactic acid bacteria, on *P. aeruginosa* and *E. coli*.

## 2. Materials and Methods

### 2.1. Extraction of Biosurfactants from CSW

The extraction of extracellular biosurfactants from CSW, provided by FeedStimulants (Reg. No. NL214247/ Lot NL-2728DK 7, Zoetermeer, The Netherlands) was performed following the protocol described in previous studies [9]. Solids content of CSW was determined by weighing a sample before and after drying at 100 °C during 48 h in a stove. CSW was diluted in distilled water up to 50 g L^−1^ and extracted with ethyl acetate (CSW solution:ethyl acetate, 1:3, *v/v*) at room temperature for 60 min. Then, the organic solvent was evaporated by vacuum distillation, obtaining a multifunctional biosurfactant extract. This non-filtered biosurfactant (NFBS) extract was then dissolved in distilled water at different concentrations in order to be characterized and to evaluate its antimicrobial capacity against pathogenic strains. All the process was carried out under sterile conditions. The extractive yield of the biosurfactant extraction from CSW with ethyl acetate has been evaluated gravimetrically following the protocol established in a previous work [2] by weighing a sample before and after drying at 100 °C during 48 h in a stove.

Likewise, in order to study the effect of filtration on the antimicrobial capacity of the biosurfactant extract obtained from corn, this was filtered with a 0.22 μm Polyvinylidene Fluoride (PVDF) Stericup^®^ 150 mL Durapore^®^ membrane (EMD Millipore Corporation, Billerica, MA, USA), and the extract obtained was named filtered biosurfactant (FBS).

### 2.2. Surface-Active Properties Characterization of Biosurfactants Extracted from CSW

The biosurfactant extract obtained from CSW was subjected to various analysis to determine the surface tension (ST) and critical micellar concentration (CMC) using a Krüss K20 EasyDyne tensiometer with a 1.9 cm platinum Willhemly plate at room temperature. Several dilutions were prepared to determine the CMC of both NFBS and FBS extracts, from CSW. All measurements were conducted by triplicate at room temperature.

### 2.3. Fourier-Transform Infrared Spectroscopy (FTIR) Characterization of Biosurfactants Extracted from CSW

NFBS and FBS extracts from CSW (1 mg) were ground and pressed with 10 mg potassium bromide (7500 kg for 30 s) to obtain translucent pellets. Infrared absorption spectra of both biosurfactant extracts obtained with ethyl acetate from CSW were recorded on a Nicolet 6700 FTIR system (Thermo Scientific) with a spectral resolution of 4 cm^−1^ and wavenumber accuracy between 400 and 4000 cm^−1^. As a background reference, a potassium bromide pellet was used in the evaluation of all measurements, obtaining 32 scans per spectrum.

### 2.4. Strains and Standard Culture Conditions for Antimicrobial Assay

The antimicrobial activity of the biosurfactant extracts obtained from CSW was assessed against two pathogenic strains obtained from the Spanish Type Culture Collection (CECT) (Valencia, Spain). The following strains were selected: *Pseudomonas aeruginosa* CECT-111 (ATCC-9027) and *Escherichia coli* CECT-516 (ATCC-8739). These strains were cultivated in Trypticase Soy Broth (TSB) medium at 37 °C for 24 h in aerobic conditions. The composition of TSB medium was: 17 g/L casein peptone, 3 g/L soy peptone, 5 g/L sodium chloride, 2.5 g/L dipotassium phosphate and 2.5 g/L dextrose.

### 2.5. Antimicrobial Assay

The antimicrobial activity of the biosurfactant extract obtained from CSW against two pathogenic strains of *P. aeruginosa* and *E. coli* was determined by measuring the absorbance at 600 nm in 96-well plates in a microplate reader (MultiSkan GO Microplate Photometer, ThermoFisher Scientific, Waltham, MA, USA) following the methodology described in recent publications [16]. Briefly, TSB medium containing different concentrations of the NFBS and FBS extracts from CSW were prepared in sterile tubes with a final volume of 3 mL. An inoculum of 30 μL of the selected pathogen, with 2 × 10^6^ CFU/mL, was used for each experiment. All samples were prepared by triplicate. Likewise, positive control consisted of TSB medium containing the pathogenic strain, in absence of biosurfactant extract; whereas negative control was formulated with TSB medium in absence of pathogenic strain. Every tube was then rinsed and incubated at 37 °C.

After 24 and 48 h, 250 μL of samples and controls were placed into the columns of 96-well microplates and measured the absorbance at 600 nm.

Growth inhibitions percentages at different diluted concentrations of biosurfactant for each pathogenic strain were calculated following Equation (1):(1)% Growth Inhibition=[1−(AcA0)] × 100
where A_c_ represents the absorbance of samples with a specific concentration of biosurfactant and A_0_ the absorbance of the positive control well.

The minimum inhibitory concentration (MIC) was determined for each of the two pathogenic strains as the lowest concentration of biosurfactant that completely inhibits growth (A_600_ = 0).

## 3. Results and Discussion

### 3.1. Biosurfactants Characterization

Biosurfactants extracted from CSW, which shows a content of 50% in solids, are secondary metabolites produced by microorganisms, including probiotic lactic acid bacteria, that grow spontaneously in this agroindustrial stream, and they have showed promising results for their application in the cosmetic, pharmaceutical or food industry as it has been proved in recent publications [1,2,3]. Extractive yields of the biosurfactant extraction with ethyl acetate resulted in values of 1%. Concerning the food industry, López-Prieto et al. [3] have demonstrated that this extract promotes the growth of probiotic bacteria like *L. casei* in drinkable yogurts, although its antimicrobial activity against pathogenic bacteria has not been evaluated at the moment.

Regarding their surface active properties, NFBS extract was able to reduce the ST of water to a minimum of 37.3 mN/m, while the FBS extract reduced the ST of water to a minimum of 44.3 mN/m, which was in the range of the results obtained in previous studies for biosurfactants extracted from CSW [5,9]. However, it is important to highlight that the lowest ST achieved with NFBS, can be due to the presence of non-soluble substances present in the extract. Hence, Figure 1, shows the variation of ST with the concentration of different aqueous solution of NFBS and FBS, observing that with NFBS was difficult to achieve a stable ST even at concentrations of 1 g/L, thus it was difficult, in this case, to calculate the CMC of NFBS (Figure 1a); whereas FBS gave a stable ST at concentrations about 300 mg/L being the CMC obtained from Figure 1b of 307 mg/L.

Additionally, Fourier-transform infrared spectroscopy (FTIR) was used to determine the possible similarities and differences between the NFBS and FBS extracted from CSW. The spectra of the NFBS and FBS are showed in Figure 2. It can be observed a strong similarity between both extracts, although FBS showed higher intensity in the band from 3500 to 3100 cm^−1^ as the result of N-H and O-H stretching indicative of amine and hydroxyl groups. Moreover, in the band from 3000 to 2800 cm^−1^ it can be observed the same intensity for both extracts, indicating the presence of aliphatic chains. In addition, a strong absorption for both extracts was observed also in the band from 1800 to 1600 cm^−1^, resulting of C=O stretching. The C-O ester groups were also confirmed by the presence of various bands from 1300 to 1000 cm^−1^, which was in concordance with previous publications [3].

### 3.2. Antimicrobial Activity

The antimicrobial activity of the extracellular biosurfactant extracted from CSW before and after a filtration stage was assessed by measuring the growth inhibition percentages obtained against two pathogenic strains of *P. aeruginosa* and *E. coli* after 24 and 48 h of incubation at 37 °C. The antimicrobial efficiency was evaluated for the NFBS and FBS extracts from CSW at different concentrations. Figure 3 and Figure 4 show the antimicrobial activities of the NFBS and FBS extracts against *P. aeruginosa* and *E. coli*, respectively. The biosurfactant extract was effective against both pathogenic strains tested at different concentrations. Comparing Figure 3a,b, it was observed that FBS extract exhibited a higher antimicrobial activity against *P. aeruginosa* than the NFBS, being able to achieve a complete growth inhibition (100%) at concentrations of 0.9 mg/mL; whereas for the NFBS, concentrations of 1.8 mg/mL only resulted in 65% of growth inhibition after 48 h (Figure 3a). Therefore, for NFBS, higher concentrations (3.5 g/L) were evaluated in order to achieve a stronger antimicrobial effect, resulting in growth inhibitions of 72% for *P. aeruginosa*. Moreover 0.5 mg/mL of FBS produced a higher inhibition after 24 h than 48 h, against *P. aeruginosa*, almost achieving a bacteriostatic effect.

As it was observed, the filtration of the biosurfactant extract supposes an important step for improving the antimicrobial capacity of the biosurfactant extract under evaluation against *P. aeruginosa*, probably due to the removal of some impurities that protect the microorganism against the active principles found in this biosurfactant extract, like phenolic compounds and lipopeptides.

On the other hand, Figure 4 shows the antimicrobial capacity of NFBS and FBS against *E coli*. Therefore, in Figure 4a it can be observed that concentrations of 2 mg/mL of NFBS inhibited completely the growth of *E. coli*, achieving better results than against *P. aeruginosa*. Although, FBS (Figure 4b) exhibited a high inhibitory effect than NFBS. Therefore, FBS at concentrations of 0.5 mg/mL, after 48 h inhibited 75% of *E. coli* growth, reaching 100% growth inhibition at concentrations of 1 mg/mL.

Regarding the results obtained with *E. coli*, it was proved that, the biosurfactant was highly more effective when it was filtered, similarly to the results obtained with *P. aeruginosa*. For the application of NFBS as antimicrobial agent, higher concentrations of biosurfactant extract have to be used.

Hence, FBS, at concentration of 1 mg/mL, possess an important bactericide capacity against *P. aeruginosa* and *E. coli,* which can be present in fruits and vegetables; whereas NFBS was only effective, in the range of concentrations tested, against *E. coli,* but at concentrations of 2 mg/mL.

For the last decade, several publications have reported the antimicrobial efficiency of surfactants of microbial origin against a wide range of pathogens. Table 1 summarizes the antimicrobial activities of biosurfactants obtained in some publications against *P. aeruginosa* and *E. coli*. It has been reported by López-Prieto et al. [24] that the microorganism responsible for the production of biosurfactants in CSW was identified and characterized as a *Bacillus* strain. As it can be observed in Table 1, several species of *Bacillus* can produce biosurfactants that show high efficiency in the inhibition of growth of *P. aeruginosa* and *E. coli*. However, the biosurfactant extracted from CSW showed more efficiency, with minimum inhibitory doses, against *P. aeruginosa* and *E. coli* than other biosurfactants reported in previous publications, where concentrations of biosurfactants extracted from *Bacillus* spp., such as *L. pentosus* and *L. paracasei*, of 25 and 50 mg/mL were used in order to achieve complete growth inhibition [13,16] as it is shown in Table 1. Likewise, biosurfactants extracted from *L**actobacillus jensenii* and *L**actobacillus rhamnosus* also inhibited the growth of several strains of *E. coli*, being close to achieve complete inhibition at concentrations of 50 mg/mL [25], although these concentrations are higher than those used with NFBS and FBS extracts from CSW. Sharma et al. [26] reported that lower concentrations of biosurfactants extracted from *L**actobacillus helveticus,* from 3 to 5 mg/mL, were able to inhibit growth of *P. aeruginosa* and *E. coli* in more than 50% as it can be observed in Table 1. Nevertheless, in comparison with the results obtained with the NFBS and FBS extracts, those concentrations resulted to be higher and less effective than the biosurfactants extracted from CSW against *P. aeruginosa* and *E. coli*, respectively, as it is showed in Figure 3 and Figure 4.

Other authors have evaluated the antimicrobial activity of biosurfactants, against *P. aeruginosa* and *E. coli*, produced by other microbial strains, different from *Bacillus*. For instance, Garg et al. [27] showed that a biosurfactant extract produced from *Candida parapsilosis* was able to inhibit 58% of growth of *E. coli* at concentrations of 5 mg/mL as it can be observed in Table 1. Although it showed to be less effective than the biosurfactant extract obtained from CSW. In addition, biosurfactants extracted from *Rhodococcus fascians* BD8 have been studied in order to assess their antimicrobial activity against *E. coli* [28], though it was not able to achieve the minimum inhibitory concentration of 50%.

The comparison of NFBS and FBS extracts with other biosurfactants studied in the literature revealed that these possess a better antimicrobial effect against pathogenic bacteria, regularly found in food products like fruits and vegetables, among others.

## 4. Conclusions

The results obtained in this work proved that the biosurfactant extract obtained from corn steep water inhibits the growth of pathogenic bacteria like *P. aeruginosa* and *E. coli*, usually found to be responsible for food spoilage in the agrifood industry. Therefore, the biosurfactant extract under evaluation could be considered as a multifunctional ingredient in the food industry. In addition, it was observed that the purification of the biosurfactant extract with PVDF membranes increased its antimicrobial activity in high extend, in comparison with the raw biosurfactant extract. Finally, taking into account that the biosurfactant under evaluation is obtained from a secondary raw material of food industry, and it is neither toxic and nor harmful for animals or humans, it could be incorporated in the agrifood industry positively, reducing the use of chemical pesticides and preservatives.

## Figures and Tables

**Figure 1 foods-08-00410-f001:**
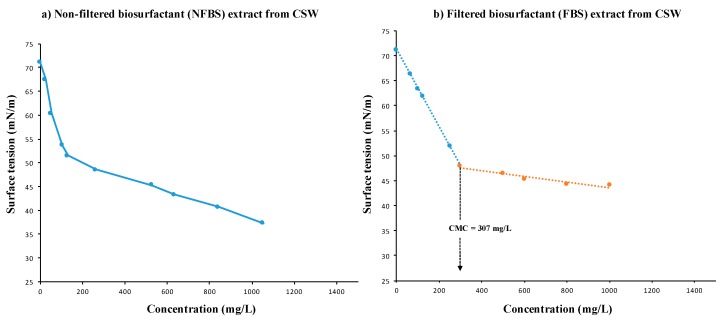
Relationship between the surface tension and biosurfactant concentration (**a**) before (NFBS) and (**b**) after (FBS) a filtration stage. CMC, Critical micellar concentration; CSW, corn steep water.

**Figure 2 foods-08-00410-f002:**
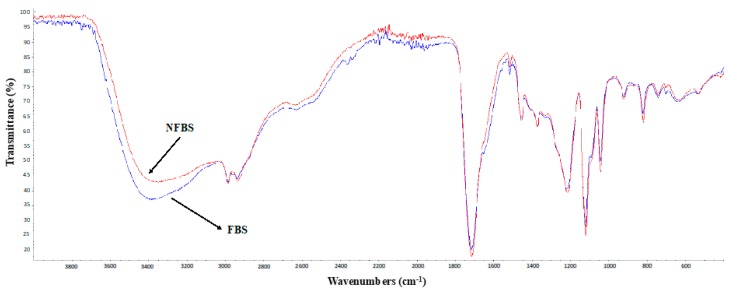
Comparison of the Fourier-Transform Infrared Spectroscopy (FTIR) spectra of the biosurfactant extracted with ethyl acetate from corn steep water (CSW) before (NFBS, red line) and after (FBS, blue line) a filtration stage.

**Figure 3 foods-08-00410-f003:**
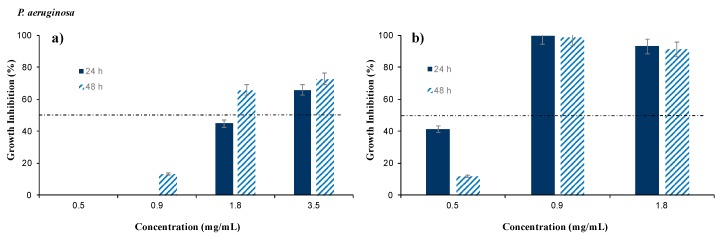
Antimicrobial activity of the biosurfactant extracted from corn steep water (CSW) (**a**) before a filtration stage (NFBS) and (**b**) after a filtration stage (FBS) against *P. aeruginosa* after 24 and 48 h of incubation. The results represent the average of triplicate experiments ± standard deviation.

**Figure 4 foods-08-00410-f004:**
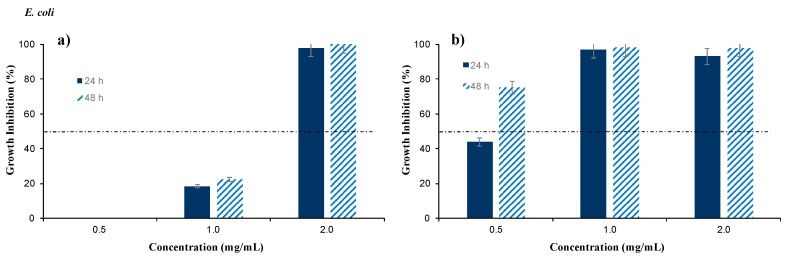
Antimicrobial activity of the biosurfactant extracted from corn steep water (CSW) (**a**) before a filtration stage (NFBS) and (**b**) after a filtration stage (FBS) against *E. coli* after 24 and 48 h of incubation. The results represent the average of triplicate experiments ± standard deviation.

**Table 1 foods-08-00410-t001:** Comparison of antimicrobial activity of different biosurfactants from the literature against *P. aeruginosa* and *E. coli* expressed in percentages of growth inhibition (PBS—Phosphate buffer saline; PB—Phosphate buffer).

Biosurfactant Source	Type of Biosurfactant	Pathogenic Strain	% of Growth Inhibition	Biosurfactant Concentration (mg/mL)	References
*Candida parapsilosis*	Extracellular	*E. coli*	58	5	Garg et al. [27]
*Lactobacillus paracasei ssp. paracasei* A20	Cell-bound	*E. coli*	100	25	Gudiña et al. [13]
		*P. aeruginosa*	91.5	50	
*Rhodococcus fascians* BD8	Extracellular	*E. coli 17-2*	25	0.5	Janek et al. [28]
		*E. coli* ATCC 10536	25	0.5	
		*E. coli* ATCC 25922	11	0.5	
*Lactobacillus jensenii*	Cell-bound	*E. coli* 438	99.0	50	Sambanthamoorthy et al. [25]
		*E. coli* 433	99.0	50	
*Lactobacillus rhamnosus*	Cell-bound	*E. coli* 438	72.34	50	
		*E. coli* 433	85.34	50	
*Lactobacillus helveticus*	Cell-bound	*E. coli* ATCC 25922	51	3.12	Sharma et al. [26]
		*P. aeruginosa* ATCC 15442	55.1	6.25	
*Lactobacillus pentosus*	Cell-bound (Extraction with PBS)	*E. coli*	89	50	Vecino et al. [16]
	Cell-bound (Extraction with PB)	*E. coli*	72	50	
	Cell-bound (Extraction with PBS)	*P. aeruginosa*	100	50	
	Cell-bound (Extraction with PB)	*P. aeruginosa*	85	50	
*Lactobacillus paracasei*	Cell-bound (Extraction with PBS)	*E. coli*	100	50	
	Cell-bound (Extraction with PB)	*E. coli*	100	50	
	Cell-bound (Extraction with PBS)	*P. aeruginosa*	100	50	
	Cell-bound (Extraction with PB)	*P. aeruginosa*	100	50	

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
