# Peer review of "A Multifunctional Biosurfactant Extract Obtained from Corn Steep Water as Bactericide for Agrifood Industry"

_foods, 2019, doi:10.3390/foods8090410_

Round 1

Reviewer 1 Report

The manuscript is in line with the topic of the Journal, focusing on the evaluation of a biosurfactant extract obtained from corn steep water and the use as bio alternative bactericide. Although sometimes the introduction revealed to be redundant and some grammar mistakes are present (check lines 34,51), the work is innovative and well done. The discussion is critical and it considers the previous literature. Some suggestions follow:

-Details about the preparation of sample CSW should be useful in materials and methods. How the content of 50% of solids has been determined?

-The quantification of extractive yield should be helpful to know if it could be applied for industrial scale-up

Author Response

The manuscript is in line with the topic of the Journal, focusing on the evaluation of a biosurfactant extract obtained from corn steep water and the use as bio alternative bactericide. Although sometimes the introduction revealed to be redundant and some grammar mistakes are present (check lines 34,51), the work is innovative and well done.

We acknowledge the positive judgment of the reviewer and grammar revisions have been made as suggested by the referee.

The discussion is critical and it considers the previous literature. Some suggestions follow:

-Details about the preparation of sample CSW should be useful in materials and methods. How the content of 50% of solids has been determined?

-The quantification of extractive yield should be helpful to know if it could be applied for industrial scale-up.

As suggested by the reviewer, details of the preparation of the sample by diluting corn steep water (CSW) in distilled water and determination of solids content gravimetrically obtaining the dry weight of the sample and quantification of extractive yield have been added to the Materials and Methods section in the paragraph referred to the extraction of biosurfactants from CSW in page 3, on lines 103 to 104 and from 109 to 112. Likewise, values of % of solids and extractive yield have been added in the Results section, page 4, lines 160, 164 and 165. All changes are highlighted in yellow in the text.

Reviewer 2 Report

The work entitled "A MULTIFUNTIONAL AND PREBIOTIC BIOSURFACTANT EXTRACT OBTAINED FROM CORN STEEP WATER AS BACTERICIDE FOR AGRIFOOD INDUSTRY "is interesting and well written.
However, I have some considerations that should be clarified. The title refers to "MULTIFUNTIONAL AND PREBIOTIC BIOSURFACTANT EXTRACT", and the conclusions indicate that "The results obtained in this work, corroborate the prebiotic character of the extract obtained from corn steep water, as it inhibits the growth of pathogenic bacteria and promotes the growth of probiotics".
The prebiotic effect refers to the stimulation of different bacterial groups at the intestinal level, as well as to the production of SCFA. This work does not provide information on this, in addition the data of the growth of probiotic bacteria are bibliographic data and are not obtained in the present work.

In addition, only lactobacillus were evaluated, it is necessary to incorporate data with bifidobacteria in order to establish more appropriate conclusions regarding the effect of these compounds on probiotic bacteria.

It is also recommended to do additional experimental work to evaluate the effect of these biocomposites using gut microbiota.

Author Response

The work entitled "A MULTIFUNTIONAL AND PREBIOTIC BIOSURFACTANT EXTRACT OBTAINED FROM CORN STEEP WATER AS BACTERICIDE FOR AGRIFOOD INDUSTRY "is interesting and well written.

We acknowledge the positive judgment of the reviewer.

However, I have some considerations that should be clarified. The title refers to "MULTIFUNTIONAL AND PREBIOTIC BIOSURFACTANT EXTRACT", and the conclusions indicate that "The results obtained in this work, corroborate the prebiotic character of the extract obtained from corn steep water, as it inhibits the growth of pathogenic bacteria and promotes the growth of probiotics".

The prebiotic effect refers to the stimulation of different bacterial groups at the intestinal level, as well as to the production of SCFA. This work does not provide information on this, in addition the data of the growth of probiotic bacteria are bibliographic data and are not obtained in the present work.

In addition, only lactobacillus was evaluated, it is necessary to incorporate data with bifidobacteria in order to establish more appropriate conclusions regarding the effect of these compounds on probiotic bacteria.

It is also recommended to do additional experimental work to evaluate the effect of these biocomposites using gut microbiota.

We acknowledge the comments of the reviewer about the current work of a biosurfactant extract from corn steep water (CSW). As correctly indicated by the referee, in this study, any experiment evaluating the prebiotic characteristics of the biosurfactant has been done. This results from the fact that in previous studies it has been proved that the biosurfactant extract from CSW promoted the growth of Lactobacillus caseicontained in a drinkable yogurt (López-Prieto et al., 2019) as it has been indicated in the text. It is well known that a good prebiotic not only promotes the growth of probiotic bacteria but also able to inhibit the growth of pathogenic strains, which in the case of the biosurfactant extracted from CSW, had not been demonstrated until this work. The current work was based on the study of the bactericide capacity of the biosurfactant under evaluation on pathogenic bacteria found in spoiled food products like Pseudomonas aeruginosa and Escherichia coli, what would complement the previous study and strengthen the prebiotic character of this biosurfactant extract. Therefore, no more assays were carried out in this work with probiotic bacteria since it was tested in previous studies (López-Prieto et al., 2019). Thus, the conclusions sections have been revised as properly commented by the reviewer. Moreover, we have removed from the manuscript those sentences related with the prebiotic character of this extract as it was not studied in the current work. Also the word "prebiotic" was removed from the title. All changes are highlighted in yellow in the text. 

Round 2

Reviewer 2 Report

no more comments